# Towards Personalized Federated Learning via Heterogeneous Model Reassembly

**Jiaqi Wang[1]    Xingyi Yang[2]    Suhan Cui[1]    Liwei Che[1]**
**Lingjuan Lyu[3]    Dongkuan Xu[4]    Fenglong Ma[1]\***
[1]The Pennsylvania State University    [2]National University of Singapore
[3]Sony AI    [4]North Carolina State University
{jqwang,sxc6192,lfc5481,fenglong}@psu.edu,
xyang@u.nus.edu, lingjuan.lv@sony.com, dxu27@ncsu.edu

## Abstract

This paper focuses on addressing the practical yet challenging problem of model heterogeneity in federated learning, where clients possess models with different network structures. To track this problem, we propose a novel framework called `pFedHR`, which leverages heterogeneous model reassembly to achieve personalized federated learning. In particular, we approach the problem of heterogeneous model personalization as a model-matching optimization task on the server side. Moreover, `pFedHR` automatically and dynamically generates informative and diverse personalized candidates with minimal human intervention. Furthermore, our proposed heterogeneous model reassembly technique mitigates the adverse impact introduced by using public data with different distributions from the client data to a certain extent. Experimental results demonstrate that `pFedHR` outperforms baselines on three datasets under both IID and Non-IID settings. Additionally, `pFedHR` effectively reduces the adverse impact of using different public data and dynamically generates diverse personalized models in an automated manner[2].

## 1    Introduction

Federated learning (FL) aims to enable collaborative machine learning without the need to share clients' data with others, thereby upholding data privacy [1, 2, 3]. However, traditional federated learning approaches [2, 4, 5, 6, 7, 8, 9, 10, 11, 12, 13, 14, 15] typically enforce the use of an identical model structure for all clients during training. This constraint poses challenges in achieving personalized learning within the FL framework. In real-world scenarios, clients such as data centers, institutes, or companies often possess their own distinct models, which may have varying structures. Training on top of their original models should be a better solution than deploying new ones for collaborative purposes. Therefore, a practical solution lies in fostering **heterogeneous model cooperation** within FL, while preserving individual model structures. Only a few studies have attempted to address the challenging problem of heterogeneous model cooperation in FL [16, 17, 18, 19, 20], and most of them incorporate the use of a public dataset to facilitate both cooperation and personalization [17, 18, 19, 20]. However, these approaches still face several key issues:

- **Undermining personalization through consensus**: Existing methods often generate consensual side information, such as class information [17], logits [18, 21], and label-wise representations [22], using public data. This information is then exchanged and used to conduct average operations on the server, resulting in a consensus representation. However, this approach poses privacy and

---

\*Corresponding author.
[2]Source code can be found in the link `https://github.com/JackqqWang/pfedHR`

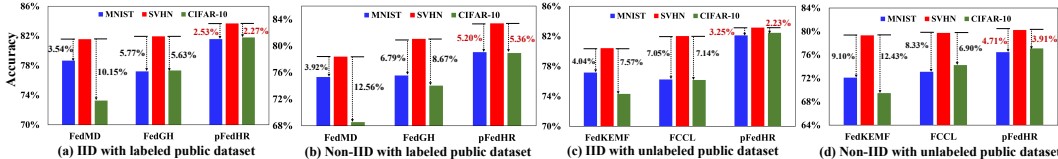

Figure 1: Performance changes when using different public data. `pFedHR` is our proposed model.

security concerns due to the exchange of side information [23]. Furthermore, the averaging process significantly diminishes the unique characteristics of individual local models, thereby hampering model personalization. Consequently, there is a need to explore approaches that can achieve local model personalization without relying on consensus-based techniques.

- **Excessive reliance on prior knowledge for distillation-based approaches**: Distillation-based techniques, such as knowledge distillation (KD), are commonly employed for heterogeneous model aggregation in FL [19, 20, 24]. However, these techniques necessitate the predefinition of a shared model structure based on prior knowledge [20]. This shared model is then downloaded to clients to guide their training process. Consequently, handcrafted models can heavily influence local model personalization. Additionally, a fixed shared model structure may be insufficient for effectively guiding personalized learning when dealing with a large number of clients with non-IID data. Thus, it is crucial to explore methods that can automatically and dynamically generate client-specific personalized models as guidance.

- **Sensitivity to the choice of public datasets**: Most existing approaches use public data to obtain guidance information, such as logits [18, 21] or a shared model [20], for local model personalization. The design of these approaches makes public data and model personalization **tightly bound together**. Thus, they usually choose the public data with the same distribution as the client data. Therefore, using public data with different distributions from client data will cause a significant performance drop in existing models. Figure 1 illustrates the performance variations of different models trained on the SVHN dataset with different public datasets (detailed experimental information can be found in Section 4.4). The figure demonstrates a significant performance drop when using alternative public datasets. Consequently, mitigating the adverse impact of employing diverse public data remains a critical yet practical research challenge in FL.

**Motivation & Challenges**. In fact, both consensus-based and distillation-based approaches aim to learn aggregated and shared information used as guidance in personalized local model training, which is not an optimal way to achieve personalization. An ideal solution is to generate a personalized model for the corresponding client, which is significantly challenging since the assessable information on the server side can only include the uploaded client models and the public data. To avoid the issue of public data sensitivity, only client models can be used. These constraints motivate us to employ the model reassembly technique [25] to generate models first and then select the most matchable personalized model for a specific client from the generations.

To this end, we will face several new challenges. (**C1**) Applying the model reassembly technique will result in many candidates. Thus, the first challenge is how to get the optimal candidates. (**C2**) The layers of the generated candidates are usually from different client models, and the output dimension size of the first layer may not align with the input dimension size of the second layer, which leads to the necessity of network layer stitching. However, the parameters of the stitched layers are unknown. Therefore, the second challenge is how to learn those unknown parameters in the stitched models. (**C3**) Even with well-trained stitched models, digging out the best match between a client model and a stitched model remains a big challenge.

**Our Approach**. To simultaneously tackle all the aforementioned challenges, we present a novel framework called `pFedHR`, which aims to achieve personalized federated learning and address the issue of heterogeneous model cooperation (as depicted in Figure 2). The `pFedHR` framework comprises two key updates: the server update and the client update. In particular, to tackle **C3**, we approach the issue of heterogeneous model personalization from a model-matching optimization perspective on the **server** side (see Section 3.1.1). To solve this problem, we introduce a novel **heterogeneous model reassembly** technique in Section 3.1.2 to assemble models uploaded from clients, i.e., $\{\mathbf{w}_t^1, \cdots, \mathbf{w}_t^B\}$, where $B$ is the number of active clients in the $t$-the communication round. This technique involves the dynamic grouping of model layers based on their functions, i.e., *layer-wise decomposition* and *function-driven layer grouping* in Figure 2. To handle **C1**, a heuristic

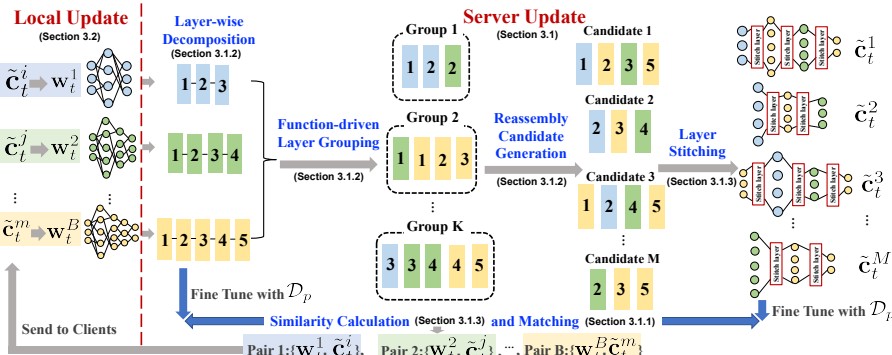

Figure 2: Overview of the proposed `pFedHR`. $K$ is the number of clusters.

rule-based search strategy is proposed in *reassembly candidate generation* to assemble informative and diverse model candidates using clustering results. Importantly, all layers in each candidate are derived from the uploaded client models.

Prior to matching the client model with candidates, we perform network *layer stitching* while maximizing the retention of information from the original client models (Section 3.1.3). To tackle **C2**, we introduce public data $\mathcal{D}_p$ to help the finetuning of the stitched candidates, i.e., $\{\tilde{\mathbf{c}}_t^1, \cdots, \tilde{\mathbf{c}}_t^M\}$, where $M$ is the number of generated candidates. Specifically, we employ labeled **OR** unlabeled public data to fine-tune the stitched and client models and then calculate similarities based on model outputs. Intuitively, if two models are highly related to each other, their outputs should also be similar. Therefore, we select the candidate with the highest similarity as the personalized model of the corresponding client, which results in matched pairs $\{\{\mathbf{w}_t^1, \tilde{\mathbf{c}}_t^i\}, \cdots, \{\mathbf{w}_t^B, \tilde{\mathbf{c}}_t^m\}\}$ in Figure 2. In the **client** update (Section 3.2), we treat the matched personalized model as a guidance mechanism for client parameter learning using knowledge distillation[3].

It is worth noting that we **minimally use public data** during our model learning to reduce their adverse impact. In our model design, the public data are used for clustering layers, fine-tuning the stitched candidates, and guiding model matching. Clustering and matching stages use public data to obtain the feedforward outputs as guidance and do not involve model parameter updates. *Only in the fine-tuning stage, the stitched models' parameters will be updated based on public data.* To reduce its impact as much as possible, we limit the number of finetuning epochs during the model implementation. Although we cannot thoroughly break the tie between model training and public data, such a design at least greatly alleviates the problem of public data sensitivity in FL.

**Contributions**. Our work makes the following key contributions: (1) We introduce the first personalized federated learning framework based on model reassembly, specifically designed to address the challenges of heterogeneous model cooperation. (2) The proposed `pFedHR` framework demonstrates the ability to automatically and dynamically generate personalized candidates that are both informative and diverse, requiring minimal human intervention. (3) We present a novel heterogeneous model reassembly technique, which effectively mitigates the adverse impact caused by using public data with distributions different from client data. (4) Experimental results show that the `pFedHR` framework achieves state-of-the-art performance on three datasets, exhibiting superior performance under both IID and Non-IID settings when compared to baselines employing labeled and unlabeled public datasets.

## 2 Related Work

**Model Heterogeneity in Federated Learning**. Although many federated learning models, such as FedAvg [4], FedProx [2], Per-FedAvg [26], PFedMe [27], and PFedBayes[28], have been proposed recently, the focus on heterogeneous model cooperation, where clients possess models with diverse structures, remains limited. It is worth noting that in this context, the client models are originally distinct and not derived from a shared, large global model through the distillation of subnetworks [29, 30]. Existing studies on heterogeneous model cooperation can be broadly categorized based on whether they utilize public data for model training. FedKD [16] aims to achieve personalized models

---

[3]Note that the network architecture of both local model and personalized model are known, and thus, there is no human intervention in the client update.

Table 1: A comparison between existing heterogeneous model cooperation works and our pFedHR.

| Approach | Public Dataset | | Model Characteristics | | |
|---|---|---|---|---|---|
| | W. Label | W.o. Label | Upload and Download | Aggregation | Personalization |
| FedDF[19] | ✗ | ✓ | parameters | ensemble distillation | ✗ |
| FedKEMF[20] | ✗ | ✓ | parameters | mutual learning | ✓ |
| FCCL [18] | ✗ | ✓ | logits | average | ✓ |
| FedMD[17] | ✓ | ✗ | class scores | average | ✓ |
| FedGH [22] | ✓ | ✗ | label-wise representations | average | ✓ |
| pFedHR | ✓ | ✓ | parameters | model reassembly | ✓ |

for each client **without employing public data** by simultaneously maintaining a large heterogeneous model and a small homogeneous model on each client, which incurs high computational costs.

Most existing approaches **leverage public data** to facilitate model training. Among them, FedDF [19], FedKEMF [20], and FCCL [18] employ *unlabeled public data*. However, FedDF trains a global model with different settings compared to our approach. FedKEMF performs mutual knowledge distillation learning on the server side to achieve model personalization, requiring predefined model structures. FCCL averages the logits provided by each client and utilizes a consensus logit as guidance during local model training. It is worth noting that the use of logits raises concerns regarding privacy and security [23, 31]. FedMD [17] and FedGH [22] employ *labeled public data*. These approaches exchange class information or representations between the server and clients and perform aggregation to address the model heterogeneity issue. However, similar to FCCL, these methods also introduce privacy leakage concerns. In contrast to existing work, we propose a general framework capable of utilizing either **labeled OR unlabeled public data** to learn personalized models through heterogeneous model reassembly. We summarize the distinctions between existing approaches and our framework in Table 1.

**Neural Network Reassembly and Stitching**. As illustrated in Table 1, conventional methods are primarily employed in existing federated learning approaches to obtain personalized client models, with limited exploration of model reassembly. Additionally, there is a lack of research investigating neural network reassembly and stitching [25, 32, 33, 34, 35, 36] within the context of federated learning. For instance, the work presented in [33] proposes three algorithms to merge two models within the weight space, but it is limited to handling only two models as input. In our setting, multiple models need to be incorporated into the model reassembly or aggregation process. Furthermore, both [25] and [34] focus on pre-trained models, which differ from our specific scenario.

## 3 Methodology

Our model pFedHR incorporates two key updates: the server update and the local update, as depicted in Figure 2. Next, we provide the details of our model design starting with the server update.

### 3.1 Server Update

During each communication round $t$, the server will receive $B$ heterogeneous client models with parameters denoted as $\{\mathbf{w}_t^1, \mathbf{w}_t^2, \cdots, \mathbf{w}_t^B\}$. As we discussed in Section 1, traditional approaches have limitations when applied in this context. To overcome these limitations and learn a personalized model $\hat{\mathbf{w}}_t^n$ that can be distributed to the corresponding $n$-th client, we propose a novel approach that leverages the publicly available data $\mathcal{D}_p$ stored on the server to find the **most similar** aggregated models learned from $\{\mathbf{w}_t^1, \mathbf{w}_t^2, \cdots, \mathbf{w}_t^B\}$ for $\mathbf{w}_t^n$.

#### 3.1.1 Similarity-based Model Matching

Let $g(\cdot, \cdot)$ denote the model aggregation function, which can automatically and dynamically obtain $M$ aggregated model candidates as follows:

$$\{\mathbf{c}_t^1, \cdots, \mathbf{c}_t^M\} = g(\{\mathbf{w}_t^1, \mathbf{w}_t^2, \cdots, \mathbf{w}_t^B\}, \mathcal{D}_p), \tag{1}$$

where $g(\cdot, \cdot)$ will be detailed in Section 3.1.2, and $\mathbf{c}_k^m$ is the $m$-th generated model candidate learned by $g(\cdot, \cdot)$. Note that $\mathbf{c}_k^m$ denotes the model before network stitching. $M$ is the total number of candidates, which is not a fixed number and is estimated by $g(\cdot, \cdot)$. In such a way, our goal is to optimize the following function:

$$\mathbf{c}_t^* = \underset{\mathbf{c}_t^m; m \in [1,M]}{\arg \max} \{\text{sim}(\mathbf{w}_t^n, \mathbf{c}_t^1; \mathcal{D}_p), \cdots, \text{sim}(\mathbf{w}_t^n, \mathbf{c}_t^M; \mathcal{D}_p)\}, \forall n \in [1, B], \tag{2}$$

---

**Algorithm 1:** Reassembly Candidate Search

**input** : Layer clusters $\{\mathcal{G}_t^1, \mathcal{G}_t^2, \cdots, \mathcal{G}_t^K\}$, operation type set $\mathcal{O}$, rule set $\mathcal{R}$
**output** : $\{\mathbf{c}_t^1, \cdots, \mathbf{c}_t^M\}$

1  Initialize $\mathcal{C}_t = \varnothing$;
2  **for** $k \leftarrow 1, \cdots, K$ **do**
3      // $Q_k$ is the number of operation-layer pairs in $\mathcal{G}_t^k$
4      **for** $q \leftarrow 1, \cdots, Q_k$ **do**
5          Initialize an empty candidate $\mathbf{c}_q = []$, operation type set $\mathcal{O}_q = \varnothing$, group id set $\mathcal{K}_q = \varnothing$;
6          Select the $q$-th layer-operation pair $(\mathcal{L}_{t,i}^n, O_i^n)$ from group $\mathcal{G}_k$
7          Add $(\mathcal{L}_{t,i}^n, O_i^n)$ to $\mathbf{c}_q$, add $O_i^n$ to $\mathcal{O}_q$, add $k$ to $\mathcal{K}_q$;
8          **for** $k' \leftarrow 1, \cdots, K$ **do**
9              Check a pair from $\mathcal{G}_{k'}$ whether it satisfies:
10             $\mathcal{R}_1$ (layer order): the layer index should be larger than that of the last layer added to $\mathbf{c}_q$, and
11             $\mathcal{R}_2$ (operation order): the operation type should be followed by the previous type in $\mathbf{c}_q$;
12             **if** *True* **then**
13                 Add the pair to $\mathbf{c}_q$, add its operation type to $\mathcal{O}_q$, add $k'$ to $\mathcal{K}_q$;
14                 Move to the next pair;
15         Check $\mathcal{O}_q$ and $\mathcal{K}_q$ with:
16         $\mathcal{R}_3$ (complete operation): the size of $\mathcal{O}_q$ should be equal to that of $\mathcal{O}$, and
17         $\mathcal{R}_4$ (diverse group): the size of $\mathcal{K}_q$ should be equal to $K$;
18         **if** *True* **then**
19             Add the candidate $\mathbf{c}_q$ to $\mathcal{C}_t$;

    **return** : $\mathcal{C}_t = \{\mathbf{c}_t^1, \cdots, \mathbf{c}_t^M\}$

---

where $\mathbf{c}_t^*$ is the best matched model for $\mathbf{w}_t^n$, which is also denoted as $\hat{\mathbf{w}}_t^n = \mathbf{c}_t^*$. $\text{sim}(\cdot, \cdot)$ is the similarity function between two models, which will be detailed in Section 3.1.3.

### 3.1.2 Heterogeneous Model Reassembly – $g(\cdot, \cdot)$

To optimize Eq. (2), we need to obtain $M$ candidates using the heterogenous model aggregation function $g(\cdot)$ in Eq. (1). To avoid the issue of predefined model architectures in the knowledge distillation approaches, we aim to automatically and dynamically learn the candidate architectures via a newly designed function $g(\cdot, \cdot)$. In particular, we propose a decomposition-grouping-reassembly method as $g(\cdot, \cdot)$, including layer-wise decomposition, function-driven layer grouping, and reassembly candidate generation.

**Layer-wise Decomposition**. Assume that each uploaded client model $\mathbf{w}_t^n$ contains $H$ layers, i.e., $\mathbf{w}_t^n = [(\mathbf{L}_{t,1}^n, O_1^n), \cdots, (\mathbf{L}_{t,H}^n, O_E^n)]$, where each layer $\mathbf{L}_{t,h}^n$ is associated with an operation type $O_e^n$. For example, a plain convolutional neural network (CNN) usually has three operations: convolution, pooling, and fully connected layers. For different client models, $H$ may be different. The decomposition step aims to obtain these layers and their corresponding operation types.

**Function-driven Layer Grouping.** After decomposing layers of client models, we group these layers based on their functional similarities. Due to the model structure heterogeneity in our setting, the dimension size of the output representations from layers by feeding the public data $\mathcal{D}_p$ to different models will be different. Thus, measuring the similarity between a pair of layers is challenging, which can be resolved by applying the commonly used centered kernel alignment (CKA) technique [37]. In particular, we define the distance metric between any pair of layers as follows:

$$\text{dis}(\mathbf{L}_{t,i}^n, \mathbf{L}_{t,j}^b) = (\text{CKA}(\mathbf{X}_{t,i}^n, \mathbf{X}_{t,j}^b) + \text{CKA}(\mathbf{L}_{t,i}^n(\mathbf{X}_{t,i}^n), \mathbf{L}_{t,i}^b(\mathbf{X}_{t,j}^b)))^{-1}, \tag{3}$$

where $\mathbf{X}_{t,i}^n$ is the input data of $\mathbf{L}_{t,i}^n$, and $\mathbf{L}_{t,i}^n(\mathbf{X}_{t,i}^n)$ denotes the output data from $\mathbf{L}_{t,i}^n$. This metric uses $\text{CKA}(\cdot, \cdot)$ to calculate the similarity between both input and output data of two layers.

Based on the defined distance metric, we conduct the K-means-style algorithm to group the layers of $B$ models into $K$ clusters. This optimization process aims to minimize the sum of distances between

all pairs of layers, denoted as $\mathcal{L}_t$. The procedure can be described as follows:

$$\min \mathcal{L}_t = \min_{\delta_{b,h}^a \in \{0,1\}} \sum_{k=1}^{K} \sum_{b=1}^{B} \sum_{h=1}^{H} \delta_{b,h}^k (\text{dis}(\mathbf{L}_t^k, \mathbf{L}_{t,h}^b)), \tag{4}$$

where $\mathbf{L}_t^k$ is the center of the $k$-th cluster. $\delta_{b,h}^k$ is the indicator. If the $h$-th layer of $\mathbf{w}_t^b$ belongs to the $k$-th cluster, then $\delta_{b,h}^k = 1$. Otherwise, $\delta_{b,h}^k = 0$. After the grouping process, we obtain $K$ layer clusters denoted as $\{\mathcal{G}_t^1, \mathcal{G}_t^2, \cdots, \mathcal{G}_t^K\}$. There are multiple layers in each group, which have similar functions. Besides, each layer is associated with an operation type.

**Reassembly Candidate Generation**. The last step for obtaining personalized candidates $\{\mathbf{c}_t^1, \cdots, \mathbf{c}_t^M\}$ is to assemble the learned layer-wise groups $\{\mathcal{G}_t^1, \mathcal{G}_t^2, \cdots, \mathcal{G}_t^K\}$ based on their functions. To this end, we design a heuristic rule-based search strategy as shown in Algorithm 1. Our goal is to automatically generate *informative* and *diverse* candidates.

Generally, an **informative** candidate needs to follow the design of handcrafted network structures. This is challenging since the candidates are automatically generated without human interventions and prior knowledge. To satisfy this condition, we require the layer orders to be guaranteed ($\mathcal{R}_1$ in Line 10). For example, the $i$-th layer from the $n$-th model, i.e., $\mathbf{L}_{t,i}^n$, in a candidate must be followed by a layer with an index $j > i$ from other models or itself. Besides, the operation type also determines the quality of a model. For a CNN model, the fully connected layer is usually used after the convolution layer, which motivates us to design the $\mathcal{R}_2$ operation order rule in Line 11.

Only taking the informativeness principle into consideration, we may generate a vast number of candidates with different sizes of layers. Some candidates may be a subset of others and even worse with low quality. Besides, the large number of candidates will increase the computational burden of the server. To avoid these issues and further obtain high-quality candidates, we use the **diversity** principle as the filtering rule. A diverse and informative model should contain all the operation types, i.e., the $\mathcal{R}_3$ complete operation rule in Line 16. Besides, the groups $\{\mathcal{G}_t^1, \cdots, \mathcal{G}_t^K\}$ are clustered based on their layer functions. The requirement that layers of candidates must be from different groups should significantly increase the diversity of model functions, which motivates us to design the $\mathcal{R}_4$ diverse group rule in Line 17.

### 3.1.3 Similarity Learning with Layer Stitching – $\text{sim}(\cdot, \cdot)$

After obtaining a set of candidate models $\{\mathbf{c}_t^1, \cdots, \mathbf{c}_t^M\}$, to optimize Eq. (2), we need to calculate the similary bettwen each client model $\mathbf{w}_t^n$ and all the cadidates $\{\mathbf{c}_t^1, \cdots, \mathbf{c}_t^M\}$ using the public data $\mathcal{D}_p$. However, this is non-trivial since $\mathbf{c}_t^m$ is assembled by layers from different client models, which is not a complete model architecture. We have to stitch these layers together before using $\mathbf{c}_t^m$.

**Layer Stitching**. Assume that $\mathbf{L}_{t,i}^n$ and $\mathbf{L}_{t,j}^b$ are any two consecutive layers in the candidate model $\mathbf{c}_t^m$. Let $d_i$ denote the output dimension of $\mathbf{L}_{t,i}^n$ and $d_j$ denote the input dimension of $\mathbf{L}_{t,j}^b$. $d_i$ is usually not equal to $d_j$. To stitch these two layers, we follow existing work [34] by adding a nonlinear activation function $\text{ReLU}(\cdot)$ on top of a linear layer, i.e., $\text{ReLU}(\mathbf{W}^\top \mathbf{X} + \mathbf{b})$, where $\mathbf{W} \in \mathbb{R}^{d_i \times d_j}$, $\mathbf{b} \in \mathbb{R}^{d_j}$, and $\mathbf{X}$ represents the output data from the first layer. In such a way, we can obtain a stitched candidate $\tilde{\mathbf{c}}_t^m$. The reasons that we apply this simple layer as the stitch are twofold. On the one hand, even adding a simple linear layer between any two consecutive layers, the model will increase $d_i * (d_j + 1)$ parameters. Since a candidate $\mathbf{c}_t^m$ may contain several layers, if using more complicated layers as the stitch, the number of parameters will significantly increase, which makes the new candidate model hard to be trained. On the other hand, using a simple layer with a few parameters may be helpful for the new candidate model to maintain more information from the original models. This is of importance for the similarity calculation in the next step.

**Similarity Calculation**. We propose to use the cosine score $\cos(\cdot, \cdot)$ to calculate the similarity between a pair of models $(\mathbf{w}_t^n, \tilde{\mathbf{c}}_t^m)$ as follows:

$$\text{sim}(\mathbf{w}_t^n, \mathbf{c}_t^m; \mathcal{D}_p) = \text{sim}(\mathbf{w}_t^n, \tilde{\mathbf{c}}_t^m; \mathcal{D}_p) = \frac{1}{P} \sum_{p=1}^{P} \cos(\boldsymbol{\alpha}_t^n(\mathbf{x}_p), \boldsymbol{\alpha}_t^m(\mathbf{x}_p)), \tag{5}$$

where $P$ denotes the number of data in the public dataset $\mathcal{D}_p$ and $\mathbf{x}_p$ is the $p$-th data in $\mathcal{D}_p$. $\boldsymbol{\alpha}_t^n(\mathbf{x}_p)$ and $\boldsymbol{\alpha}_t^m(\mathbf{x}_p)$ are the logits output from models $\mathbf{w}_t^n$ and $\tilde{\mathbf{c}}_t^m$, respectively. To obtain the logits, we

need to finetune $\mathbf{w}_t^n$ and $\tilde{\mathbf{c}}_t^m$ using $\mathcal{D}_p$ first. In our design, we can use both labeled and unlabeled data to finetune models but with different loss functions. If $\mathcal{D}_p$ is **labeled**, then we use the supervised cross-entropy (CE) loss to finetune the model. If $\mathcal{D}_p$ is **unlabeled**, then we apply the self-supervised contrastive loss to finetune them following [38].

## 3.2 Client Update

The obtained personalized model $\hat{\mathbf{w}}_t^n$ (i.e., $\mathbf{c}_t^*$ in Eq. (2)) will be distributed to the $n$-th client if it is selected in the next communication round $t + 1$. $\hat{\mathbf{w}}_t^n$ is a reassembled model that carries external knowledge from other clients, but its network structure is different from the original $\mathbf{w}_t^n$. To incorporate the new knowledge without training $\mathbf{w}_t^n$ from scratch, we propose to apply knowledge distillation on the client following [39].

Let $\mathcal{D}_n = \{(\mathbf{x}_i^n, \mathbf{y}_i^n)\}$ denote the labeled data, where $\mathbf{x}_i^n$ is the data feature and $\mathbf{y}_i^n$ is the coresponding ground truth vector. The loss of training local model with knowledge distillation is defined as follows:

$$\mathcal{J}_n = \frac{1}{|\mathcal{D}_n|} \sum_{i=1}^{|\mathcal{D}_n|} \left[ \text{CE}(\mathbf{w}_t^n(\mathbf{x}_i^n), \mathbf{y}_i^n) + \lambda \text{KL}(\boldsymbol{\alpha}_t^n(\mathbf{x}_i^n), \hat{\boldsymbol{\alpha}}_t^n(\mathbf{x}_i^n)) \right], \tag{6}$$

where $|\mathcal{D}_n|$ denotes the number of data in $\mathcal{D}_n$, $\mathbf{w}_t^n(\mathbf{x}_i^n)$ means the predicted label distribution, $\lambda$ is a hyperparameter, $\text{KL}(\cdot, \cdot)$ is the Kullback–Leibler divergence, and $\boldsymbol{\alpha}_t^n(\mathbf{x}_i^n)$ and $\hat{\boldsymbol{\alpha}}_t^n(\mathbf{x}_i^n)$ are the logits from the local model $\mathbf{w}_t^n$ and the downloaded personalized model $\hat{\mathbf{w}}_t^n$, respevtively.

# 4 Experiments

## 4.1 Experiment Setups

**Datasets**. We conduct experiments for the image classification task on MNIST, SVHN, and CIFAR-10 datasets under both IID and non-IID data distribution settings, respectively. We split the datasets into 80% for training and 20% for testing. During training, we randomly sample 10% training data to put in the server as $\mathcal{D}_p$ and the remaining 90% to distribute to the clients. The training and testing datasets are randomly sampled for the IID setting. For the non-IID setting, each client randomly holds two classes of data. To test the personalization effectiveness, we sample the testing dataset following the label distribution as the training dataset. We also conduct experiments to test models using **different public data** in Section 4.4.

**Baselines**. We compare pFedHR with the baselines under two settings. (1) **Heterogenous setting**. In this setting, clients are allowed to have different model structures. The proposed pFedHR is general and can use both labeled and unlabeled public datasets to conduct heterogeneous model cooperation. To make fair comparisons, we use FedMD [17] and FedGH [22] as baselines when using the *labeled public data*, and FCCL [18] and FedKEMF [20] when testing the *unlabeled public data*. (2) **Homogenous setting**. In this setting, clients share an identical model structure. We use traditional and personalized FL models as baselines, including FedAvg [4], FedProx [2], Per-FedAvg [26], PFedMe [27], and PFedBayes [28].

## 4.2 Heterogenous Setting Evaluation

**Small Number of Clients**. Similar to existing work [18], to test the performance with a small number of clients, we set the client number $N = 12$ and active client number $B = 4$ in each communication round. We design 4 types of models with different structures and randomly assign each type of model to 3 clients. The *Conv* operation contains convolution, max pooling, batch normalization, and ReLu, and the *FC* layer contains fully connected mapping, ReLU, and dropout. We set the number of clusters $K = 4$. Then local training epoch and the server finetuning epoch are equal to 10 and 3, respectively. The public data and client data are from the same dataset.

Table 2 shows the experimental results for the heterogeneous setting using both labeled and unlabeled public data. We can observe that the proposed pFedHR achieves state-of-the-art performance on all datasets and settings. We also find the methods using labeled public datasets can boost the performance compared with unlabeled public ones in general, which aligns with our expectations and experiences.

Table 2: Performance comparison with baselines under the heterogeneous setting.

| Public Data | Dataset | MNIST | | SVHN | | CIFAR-10 | |
|---|---|---|---|---|---|---|---|
| | Model | IID | Non-IID | IID | Non-IID | IID | Non-IID |
| Labeled | FedMD [17] | 93.08% | 91.44% | 81.55% | 78.39% | 68.22% | 66.13% |
| | FedGH [22] | 94.10% | 93.27% | 81.94% | 81.06% | 72.69% | 70.27% |
| | pFedHR | 94.55% | 94.41% | 83.68% | 83.40% | 73.88% | 71.74% |
| Unlabeled | FedKEMF [20] | 93.01% | 91.66% | 80.41% | 79.33% | 67.12% | 66.93% |
| | FCCL [18] | 93.62% | 92.88% | 82.03% | 79.75% | 68.77% | 66.49% |
| | pFedHR | 93.89% | 93.76% | 83.15% | 80.24% | 69.38% | 68.01% |

**Large Number of Clients**. We also test the performance of models with a large number of clients. When the number of active clients $B$ is large, calculating layer pair-wise distance values using Eq. (3) will be highly time-consuming. To avoid this issue, a straightforward solution is to conduct FedAvg [4] for averaging the models with the same structures first and then do the function-driven layer grouping based on the averaged structures via Eq. (4). The following operations are the same as pFedHR. In this experiment, we set $N = 100$ clients and the active number of clients $B = 10$. Other settings are the same as those in the small number client experiment.

Table 3 shows the results on the SVHN dataset for testing the proposed pFedHR for a large number of clients setting. The results show similar patterns as those listed in Table 2, where the proposed pFedHR achieves the best performance under IID and Non-IID settings whether it uses labeled or unlabeled public datasets. Compared to the results on the SVHN dataset in Table 2, we can find that the performance of all the baselines and our models drops. Because the number of training data is fixed, allocating these

Table 3: Evaluation using a large number of clients on the SVHN dataset ($N = 100$).

| Public Data | Model | IID | Non-IID |
|---|---|---|---|
| Labeled | FedMD | 78.16% | 74.34% |
| | FedGH | 76.27% | 72.78% |
| | pFedHR | 80.02% | 77.63% |
| Unlabeled | FedKEAF | 76.27% | 74.61% |
| | FCCL | 75.03% | 71.54% |
| | pFedHR | 78.98% | 75.77% |

data to 100 clients will make each client use fewer data for training, which leads to a performance drop. The results on both small and large numbers of clients clearly demonstrate the effectiveness of our model for addressing the heterogeneous model cooperation issue in federated learning.

## 4.3 Homogeneous Setting Evaluation

In this experiment, all the clients use the same model structure. We test the performance of our proposed pFedHR on representative models with the smallest (**M1**) and largest (**M4**) number of layers, compared with state-of-the-art homogeneous federated learning models. Except for the identical model structure, other experimental settings are the same as those used in the scenario of the small number of clients. Note that for pFedHR, we report the results using the labeled public data in this experiment. The results are shown in Table 4.

Table 4: Homogeneous model comparison with baselines.

| Model | Dataset | MNIST | | SVHN | | CIFAR-10 | |
|---|---|---|---|---|---|---|---|
| | Setting | IID | Non-IID | IID | Non-IID | IID | Non-IID |
| M1 | FedAvg [4] | 91.23% | 90.04% | 53.45% | 51.33% | 43.05% | 33.39% |
| | FedProx [2] | 92.66% | 92.47% | 54.86% | 53.09% | 43.62% | 35.06% |
| | Per-FedAvg [26] | 93.23% | 93.04% | 54.29% | 52.04% | 44.14% | 42.02% |
| | PFedMe [27] | 93.57% | 92.00% | 55.01% | 53.78% | 45.01% | 43.65% |
| | PFedBayes [28] | 94.39% | 93.32% | 58.49% | 55.74% | 46.12% | 44.49% |
| | pFedHR | 94.26% | 93.26% | 61.72% | 59.23% | 54.38% | 48.44% |
| M4 | FedAvg [4] | 94.24% | 92.16% | 83.26% | 82.77% | 67.68% | 58.92% |
| | FedProx [2] | 94.22% | 93.22% | 84.72% | 83.00% | 71.24% | 63.98% |
| | Per-FedAvg [26] | 95.77% | 93.67% | 85.99% | 84.01% | 79.56% | 76.23% |
| | PFedMe [27] | 95.71% | 94.02% | 87.63% | 85.33% | 79.88% | 77.56% |
| | PFedBayes [28] | 95.64% | 93.23% | 88.34% | 86.28% | 80.06% | 77.93% |
| | pFedHR | 94.88% | 93.77% | 89.87% | 87.94% | 81.54% | 79.45% |

We can observe that using a simple model (M1) can make models achieve relatively high performance since the MNIST dataset is easy. However, for complicated datasets, i.e., SVHN and CIFAR-10, using a complex model structure (i.e., M4) is helpful for all approaches to improve their performance significantly. Our proposed pFedHR outperforms all baselines on these two datasets, even equipping with a simple model M1. Note that our model is proposed to address the heterogeneous model cooperation problem instead of the homogeneous personalization in FL. Thus, it is a practical

approach and can achieve personalization. Still, we also need to mention that it needs extra public data on the server, which is different from baselines.

## 4.4 Public Dataset Analysis

**Sensitivity to the Public Data Selection**. In the previous experiments, the public and client data are from the same dataset, i.e., having the same distribution. To validate the effect of using different public data during model learning for all baselines and our model, we conduct experiments by choosing public data from different datasets and report the results on the SVHN dataset. Other experimental settings are the same as those in the scenario of the small number of clients.

Figure 1 shows the experimental results for all approaches using labeled and unlabeled public datasets. We can observe that replacing the public data will make all approaches decrease performance. This is reasonable since the data distributions between public and client data are different. However, compared with baselines, the proposed pFedHR has the **lowest performance drop**. *Even using other public data, pFedHR can achieve comparable or better performance with baselines using SVHN as the public data*. This advantage stems from our model design. As described in Section 3.1.3, we keep more information from original client models by using a simple layer as the stitch. Besides, we aim to search for the most similar personalized candidate with a client model. We propose to calculate the average logits in Eq. (5) as the criteria. To obtain the logits, we do not need to finetune the models many times. In our experiments, we set the number of finetuning epochs as 3. This strategy can also help the model reduce the adverse impact of public data during model training.

**Sensitivity to the Percentage of Public Data**. Several factors can affect model performance. In this experiment, we aim to investigate whether the percentage of public data is a key factor in influencing performance change. Toward this end, we still use the small number of clients setting, i.e., the client and public data are from the SVHN dataset, but we adjust the percentage of public data. In the original experiment, we used 10% data as the public data. Now, we reduce this percentage to 2% and 5%. The results are shown in Figure 3. We can observe that with the increase in the percentage of public data, the performance of the proposed pFedHR also improves. These results align with

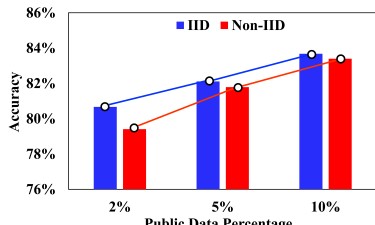

Figure 3: Performance change w.r.t. the percentage of public data.

our expectations since more public data used for finetuning can help pFedHR obtain more accurately matched personalized models, further enhancing the final accuracy.

## 4.5 Experiment results with Different Numbers of Clusters

In our model design, we need to group functional layers into $K$ groups by optimizing Eq. (4). Where $K$ is a predefined hyperparameter. In this experiment, we aim to investigate the performance influence with regard to $K$. In particular, we conduct the experiments on the SVHN dataset with 12 local clients, and the public data are also the SVHN data.

Figure 4 shows the results on both IID and Non-IID settings with labeled and unlabeled public data. $X$-axis represents the number of clusters, and $Y$-axis denotes the accuracy values. We can observe that with the increase of $K$, the performance will also increase. However, in the experiments, we do not recommend setting a large $K$ since a trade-off balance exists between $K$ and $M$, where $M$ is the number of candidates automatically generated by Algorithm 1 in the main manuscript. If $K$ is large, then $M$ will be small due to Rule $\mathcal{R}_4$. In other words, a larger $K$ may make the empty $\mathcal{C}_t$ returned by Algorithm 1 in the main manuscript.

## 4.6 Layer Stitching Study

One of our major contributions is to develop a new layer stitching strategy to reduce the adverse impacts of introducing public data, even with different distributions from the client data. Our proposed strategy includes two aspects: (1) using a simple layer to stitch layers and (2) reducing the number of finetuning epochs. To validate the correctness of these assumptions, we conduct the following experiments on SVHN with 12 clients, where both client data and **labeled** public data are extracted from SVHN.

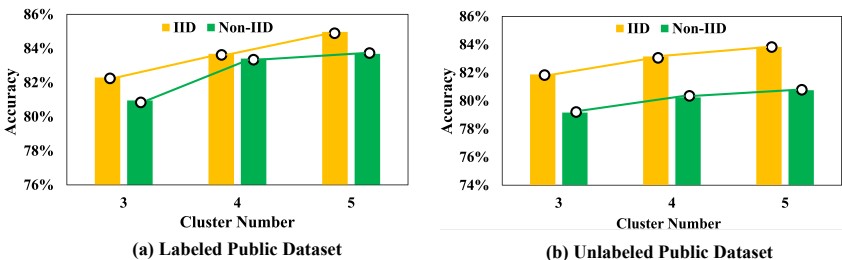

Figure 4: Results on different number of clusters $K$'s.

**Stitching Layer Numbers**. In this experiment, we add the complexity of layers for stitching. In our model design, we only use ReLU($\mathbf{W}^\top \mathbf{X} + \mathbf{b}$). Now, we increase the number of linear layers from 1 to 2 to 3. The results are depicted in Figure 5. We can observe that under both IID and Non-IID settings, the performance will decrease with the increase of the complexity of the stitching layers. These results demonstrate our assumption that more complex stitching layers will introduce more information about public data but reduce the personalized information of each client model maintained. Thus, using a simple layer to stitch layers is a reasonable choice.

**Stitched Model Finetuning Numbers**. We further explore the influence of the number of finetuning epochs on the stitched model. The results are shown in Figure 6. We can observe that increasing the number of finetuning epochs can also introduce more public data information and reduce model performance. Thus, setting a small number of finetuning epochs benefits keeping the model's performance.

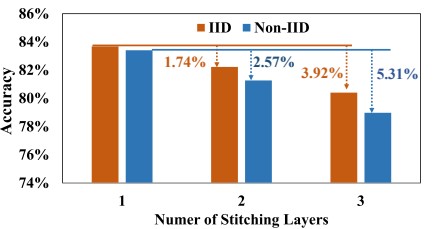

Figure 5: Stitching layer number study.

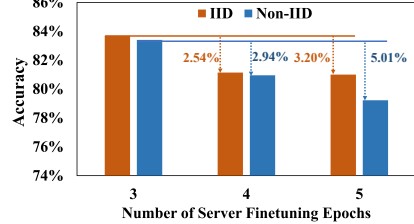

Figure 6: Server finetuning number study.

### 4.7 Personalized Model Visualization

`pFedHR` can automatically and dynamically generate candidates for clients using the proposed heterogeneous model reassembly techniques in Section 3.1.2. We visualize the generated models for a client at two different epochs ($t$ and $t'$) to make a comparison in Figure 7. We can observe that at epoch $t$, the layers are "[*Conv2* from M1, *Conv3* from M2, *Conv4* from M3, *Conv5* from M3, *FC3* from M2]", which is significantly different the model structure at epoch $t'$. These models automatically reassemble different layers from different models

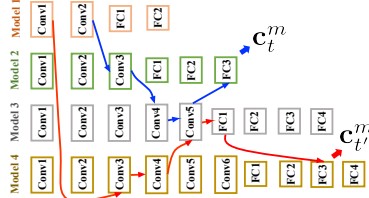

Figure 7: Candidate models.

learned by the proposed `pFedHR` instead of using predefined structures or consensus information.

## 5 Conclusion

Model heterogeneity is a crucial and practical challenge in federated learning. While a few studies have addressed this problem, existing models still encounter various issues. To bridge this research gap, we propose a novel framework, named `pFedHR`, for personalized federated learning, focusing on solving the problem of heterogeneous model cooperation. The experimental results conducted on three datasets, under both IID and Non-IID settings, have verified the effectiveness of our proposed `pFedHR` framework in addressing the model heterogeneity issue in federated learning. The achieved state-of-the-art performance serves as evidence of the efficacy and practicality of our approach.

**Acknowledgements** This work is partially supported by the National Science Foundation under Grant No. 2212323 and 2238275.

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

# Towards Personalized Federated Learning via Heterogeneous Model Reassembly
# (Appendix)

## 1 Pseudo-code of `pFedHR`

Algorithm 1 shows the pseudo-code of the proposed `pFedHR` model, which contains two main updates: the server update (lines 3-11) and the client update (lines 12 - 17). The details of reassembly candidate generation can be found in the main manuscript (Algorithm 1 on Page 5).

---

**Algorithm 1:** Algorithm Flow of `pFedHR`.

---

**Input:** Local data $\mathcal{D}_1, \mathcal{D}_2, .., \mathcal{D}_N$, the number of active clients $B$, communication rounds $T$, the number of local training epochs $E_c$, the number of fine-tuning epochs at the server $E_s$, cluster number $K$.

1   **for** each communication round $t = 1, 2, \cdots, T$ **do**
2     Randomly sample $B$ active clients and their model parameters are denoted as $\{\mathbf{w}_t^1, \mathbf{w}_t^2, \cdots, \mathbf{w}_t^B\}$;
3     **Server Update**
4       **for** each $n \in [1, \cdots, B]$ **do**
5         Conduct *layer-wise ecomposition* on $\mathbf{w}_t^n = [(\mathbf{L}_{t,1}^n, O_1^n), \cdots, (\mathbf{L}_{t,H}^b, O_E^n)]$;
6       **end**
7       Conduct *function-driven layer grouping* with Eq. (4) to obtain $\{\mathcal{G}_t^1, \mathcal{G}_t^2, \cdots, \mathcal{G}_t^K\}$;
8       Conduct *reassembly candidate generation* to obtain $\{\mathbf{c}_t^1, \cdots, \mathbf{c}_t^M\}$ following $\mathcal{R}_1, \mathcal{R}_2, \mathcal{R}_3,$ and $\mathcal{R}_4$;
9       Conduct *layer stitching* on the generated candidates;
10      Conduct *similarity calculation* on pairs of models with Eq. (5);
11      Distribute the personalized models back to the clients accordingly.
12     **Client Update**
13       **for** each $n \in [1, \cdots, B]$ **do**
14         **for** each local epoch $e$ from 1 to $E_c$ **do**
15           Update $\mathbf{w}_t^n$ with Eq. (6);
16         **end**
17       **end**
18     Upload $B$ models $\{\mathbf{w}_t^1, \mathbf{w}_t^2, \cdots, \mathbf{w}_t^B\}$ back to the server;
19   **end**

---

## 2 Implementation Details

The proposed `pFedHR` is implemented in Pytorch 2.0.1 and runs on NVIDIA A100 with CUDA version 12.0 on a Ubuntu 20.04.6 LTS server. The hyperparameter $\lambda$ in Eq. (6) is 0.2. We use Adam as the optimizer. The learning rate of the local client learning and the server fine-tuning learning rate are both equal to 0.001.

37th Conference on Neural Information Processing Systems (NeurIPS 2023).

## 2.1 Model Details

In our experiments, we have 4 CNN models with different complexity. The details are shown as follows. In each convolutional NN sequential block, there are 1 convolutional layer, a max pooling layer, and a ReLU function.

**M1**: *Cov1:{Conv2d (kernel size = 5) → ReLU → MaxPool2D (kernel size = 2, stride = 2) } → Cov2:{Conv2d (kernel size = 5) → ReLU → MaxPool2D (kernel size = 2, stride = 2) } → FC1:{Linear → ReLU} → Dropout→ FC2:Linear.*

**M2**: *Cov1:{Conv2d (kernel size = 5) → ReLU → MaxPool2D (kernel size = 2, stride = 2) } → Cov2:{Conv2d (kernel size = 5) → ReLU → MaxPool2D (kernel size = 2, stride = 2) } → Cov3:{Conv2d (kernel size = 5) → ReLU} → FC1:{Linear → ReLU} → Dropout→ FC2:Linear.*

**M3**: *Cov1:{Conv2d (kernel size = 5) → ReLU → MaxPool2D (kernel size = 2, stride = 2) } → Cov2:{Conv2d (kernel size = 5) → ReLU → MaxPool2D (kernel size = 2, stride = 2) } → Cov3:{Conv2d (kernel size = 5) → ReLU} → Cov4:{Conv2d (kernel size = 5) → ReLU → MaxPool2D (kernel size = 2, stride = 2) } → Cov4:{Conv2d (kernel size = 5) → ReLU} → FC1:{Linear → ReLU → Dropout} → FC2:{Linear → ReLU}→ FC3:{Linear → ReLU}→FC4:Linear.*

**M4**: *Cov1:{Conv2d (kernel size = 5) → BatchNorm2d → ReLU} → Cov2:{Conv2d (kernel size = 3) → ReLU → MaxPool2D (kernel size = 2, stride = 2) } → Cov3:{Conv2d (kernel size = 3) → BatchNorm2d → ReLU} → Cov4:{Conv2d (kernel size = 5) → ReLU → MaxPool2D (kernel size = 2, stride = 2) → Dropout } → Cov5:{Conv2d (kernel size = 3) → BatchNorm2d → ReLU } → Cov6:{Conv2d (kernel size = 3) → ReLU → MaxPool2D (kernel size = 2, stride = 2) } → FC1:{Linear → ReLU → Dropout} → FC2:{Linear → ReLU}→ FC3:{Linear → ReLU}→FC4:Linear.*

## 2.2 Details of Rule $\mathcal{R}_2$ (Operation Order)

We introduce the rule $\mathcal{R}_2$ (operation order) in Line 11 of the algorithm in the main manuscript. We follow existing work [1, 2, 3, 4, 5] to define the following rules:

1. A CNN typically has convolutional layers, pooling layers, and fully connected layers, which requires the generated candidate to have these functional layers.

2. The typical operation order is convolution layers → ReLU layers → pooling layers → fully connected layers.

## 2.3 Public Data Sensitivity Analysis

### 2.3.1 Experiment Results of Public Data Sensitivity on SVHN with 100 Clients

Figure 1 shows the experimental results with 100 clients. The clients hold the SVHN data, and we alternatively change the (labeled and unlabeled) public data and report the results for both IID and Non-IID settings. We can observe the same results as those on the 12 clients (Figure 1 in the main manuscript). All approaches can achieve the best performance when using SVNH as the public data. Although using other public data, the performance of all approaches drops, the performance change of the proposed pFedHR is lowest. Besides, even using other public data, pFedHR can achieve comparable or better performance to baselines. For example, in Figure 1(b), when pFedHR uses the MNIST as the public data, its performance is comparable to FedMD and better than FedGH using SVHN as the public data. These results confirm that the proposed pFedHR can handle the public data sensitivity issue.

### 2.3.2 Experiment Results of Public Data Sensitivity on CIFAR-10 with 12 Clients

We also validate the public data sensitivity on CIFAR-10, where clients hold data from CIFAR-10, and we alternatively change the public data. The results are shown in Figures 2 and 3. We have the same observations as we discussed before. Using public data with different distributions makes the performance drop, but the proposed pFedHR has the smallest drop ratio.

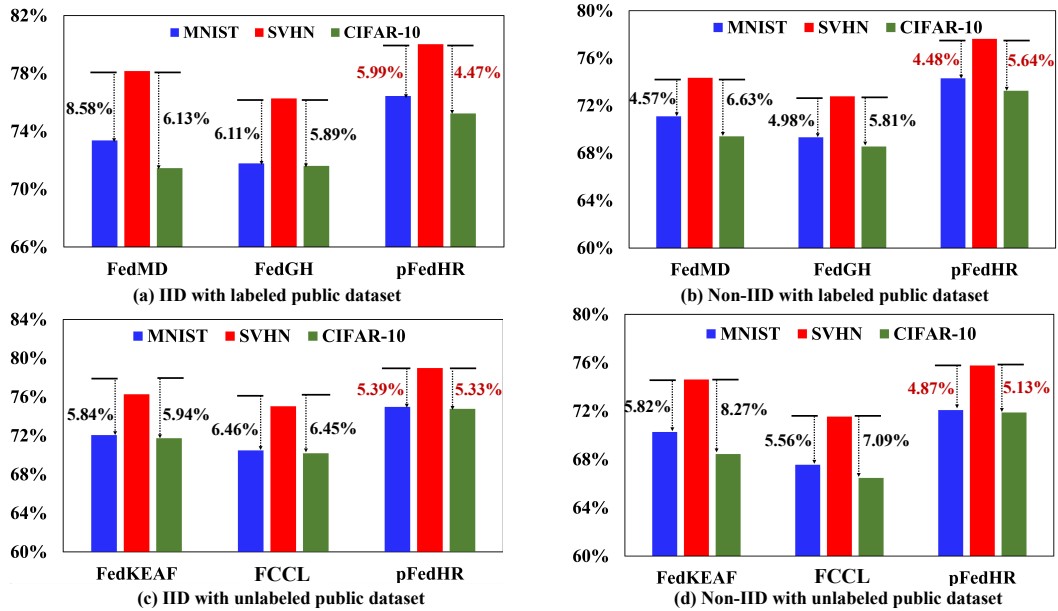

Figure 1: Results with 100 clients.

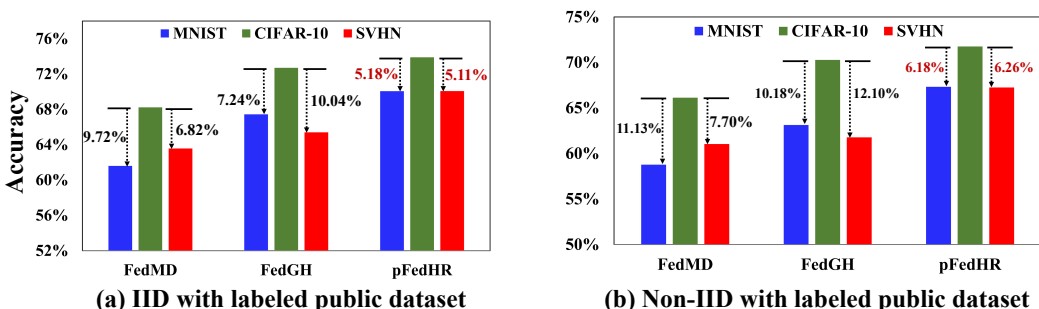

Figure 2: Results with CIFAR-10 private dataset and labeled public datasets.

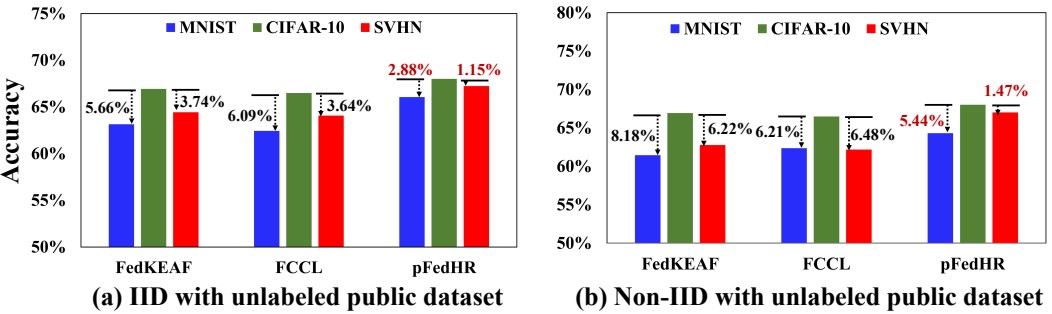

Figure 3: Results with CIFAR-10 private dataset and unlabeled public datasets.

## 3 Limitation

All the experimental results demonstrate the effectiveness of the proposed `pFedHR`. However, `pFedHR` still have the following limitations:

1. In our experiments, we use four different CNN-based models and randomly send them to clients. The reason for using these simple models is that they can save computational resources. In our model design, we will calculate the CKA score between any pair of layers.

In other words, if the local model is very deep (with multiple layers), the computational complexity will be very high.

2. In real-world applications, the client models may be more complicated. To deploy the proposed `pFedHR` in a real environment, we need to slightly modify the layer-wise decomposition and use block-wise decomposition as [6] and redefine the rules used for generating candidates accordingly.

3. In our experiments, we only test the designed model on the image datasets. In real-world applications, multiple types of data may be stored in each client. How to handle other types of data with heterogeneous model reassembly is one of our major future works.

## 4 Broader Impacts

The proposed `pFedHR` addresses the practical challenge of model heterogeneity in federated learning. By introducing heterogeneous model reassembly and personalized federated learning, this research contributes to the advancement of federated learning techniques, potentially improving the efficiency and effectiveness of collaborative machine learning in distributed systems.

The `pFedHR` framework automatically generates informative and diverse personalized candidates with minimal human intervention. This has the potential to reduce the burden on human experts and practitioners, making the process of model personalization more efficient and scalable. It opens up possibilities for deploying federated learning systems in real-world scenarios where manual customization is impractical or time-consuming.

The proposed heterogeneous model reassembly technique in `pFedHR` tries to mitigate the adverse impact caused by using public data with different distributions from the client data. This can be beneficial in scenarios where privacy concerns limit the availability of extensive client data, such as healthcare, enabling the utilization of publicly available data while maintaining a certain level of model performance. It promotes the ethical and responsible use of data and encourages collaboration between organizations and researchers in a privacy-preserving manner.

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
