# OpenReview forum: "Towards Personalized Federated Learning via Heterogeneous Model Reassembly"
_NeurIPS.cc/2023/Conference — NeurIPS 2023 poster_

### Official Review · Reviewer_xdZD · 2023-06-23

**Soundness:** 3 good
**Presentation:** 3 good
**Contribution:** 2 fair
**Rating:** 5
**Confidence:** 4

**Summary:**

The paper proposes to use the recently published model reassembly technique (NeurIPS 2022) to obtain personalized models through federated learning. At each round, the server collects the current models from the clients and uses reassembly to generate new candidate models, potentially training some stitching layers using a public datasets. Then the closest candidate model (again similarity is evaluated through the public dataset) is sent back to the client who can (later) train its own model distilling knowledge from the candidate model. By using the public dataset only for a few selected operations, the proposed scheme should be more robust to deviations from the training dataset and the public one.


**Strengths:**

* The idea to use reassembly for federated learning is a novel one to the best of my knowledge
* Experimental results are promising and the robustness to the choice of the public dataset is definitely an important plus of the proposed approach.


**Weaknesses:**

* In the proposed scheme, the server does not keep historical aggregate information about the training, as for example it does under FedAvg by storing the last version of the shared model.  Historical information is rather kept at the client, which at each round performs a local training with knowledge distillation from the candidate model selected by the server during the previous communication round to which the client participated.  As a consequence the method does not seem suited for large-scale cross-device settings where clients may be selected only a few times. The authors have considered clients' sampling rates between 1/3 and 1/10. I expect performance to decrease significantly for lower rates.
* The proposed solution requires the server to maintain the identity of the each client (to be able to send back the relevant candidate model). This prevents the applicability of privacy-preserving techniques like secure aggregation. Note that there are other personalized approaches which do not have this constraints (e.g., Ditto, FedEM,...)
* Computational overhead. If I understood correctly, the server needs to train the stitching part for every possible client/candidate-model pair, i.e., to train in total BM models, which poses a significant load on the server.
* Complexity of the proposed solution. It would have been good to perform an ablation study to evaluate if all pFedHR steps are really needed. For example, what if the clients' models are directly compared and the closest one is sent as candidate model to the client without performing any reassembly and stitching?
* The candidate model can be more complex than the client's model. There is then an implicit assumption that, while the client has selected a given model size for example on the basis of its computational and memory capabilities, it is still able to use a more complex model for knowledge distillation at training time
* The comparison with the previous literature is not always clear. Two examples:
	1. a limitation of previous literature would be that "the averaging process significantly diminishes the characteristics of individual local models" I found this sentence too vague.
	2. "however, FedDF trains a global model with different settings compared with our approach."  Again, this is too vague, what is the difference with FedDF in a few words?
* compute
	* the authors checked the compute checkbox but I was not able to find any information about computation in the paper or in the supplementary material
* reproducibility
	* while the code is provided there is no readme file about how to use it and how to reproduce the results in the paper.
* minors:
	* footnote numbers should go after punctuation marks
	* report the number of clients for table 4
	* typos: bettwen and cadidates


**Questions:**

* in the homogeneous setting, candidate model generation reduces to simply swapping layers across clients' models? or is it still possible to obtain different models' architectures (e.g. with a higher number of layers than any of the original models?)
* are results reported averages over different experiments or single-run experiments? I am also asking because some of the reported differences are really small (e.g., in table 2 pFedHR outperforms sota by less than 1 percentage point in many configurations)

**Limitations:**

I think the paper should have discussed the following limitations (see corresponding weaknesses above)
* performance under low clients' sampling rate
* the server keeps track of clients' update
* Computational overhead
* The clients need to work with models more complex than its own.

---

> ### Author Rebuttal · Authors · 2023-08-09
>
> Thanks for the reviewer's valuable comments. Hope our responses below adequately address your concerns.
>
> `>>> W1` ***Historical information maintaince***
>
> `>>> WA1`
>
> (1)  Candidate models are reassembled using the layers from the models of active clients. Our designed optimization approach enforces that the selected layers are from different functional groups and different operation sets. Though there is no single global model aggregated, the historical information is still able to be maintained.
>
> (2) We add more experiments with keeping all the settings the same as the experiment in the subsection "Large Number of Clients" of the original paper, except that we changed the sample ratio from 0.1 to ` 0.05`. With a lower active client ratio, the size candidate model pool will shrink, and the quality of the personalized model that clients obtain will be affected, which results in a performance drop.
> | Public Data | Active Client Ratio | IID     | Non-IID |
> |-------------|---------------------|---------|---------|
> | Labeled     | 0.05                | 79.21\% | 75.86\% |
> |             | 0.1                 | 80.02\% | 77.63\% |
> | Unlabeled   | 0.05                | 78.36\% | 74.70\% |
> |             | 0.1                 | 78.98\% | 75.77\% |
>
> `>>> W2` ***Privacy***
>
> `>>> WA2`
>
> (1) Thanks for the valuable suggestion. In our setting, the model structures of the clients are heterogeneous, which prevents us from using the classic model aggregation approach. Therefore, the aggregation-based privacy-preserving techniques may not be directly applied to our approach.
>
> (2) In our work, we fully disassemble models into layers, and the candidate models are automatically generated. After this step, the original model structure information would be changed. After assembly with the stitching layers, we tune the stitching layer parameters to obtain the models, which leads to the change of the model parameters. In such a way, it may alleviate privacy concerns. Designing a more secure and private methodology for our current approach would be one of the convincing future works.
>
> `>>> W3` ***Computational overhead***
>
> `>>> WA3`
>
> (1) Your understanding is correct. We only need to train the stitching parts rather than the whole model. The depth of the stitching layers is not very huge.
>
> (2) We first cluster layers into different groups based on their functionality and follow all the designed rules to get the candidate models. With such constraints, we restrict the size of the candidate model pool.
>
> (3) In most FL settings, we would assume that the server may have considerable computational resources. Tuning the stitching layers of the candidate models can be conducted in a parallel way.
>
> `>>> W4` ***Ablation study***
>
> `>>> WA4`
>
> (1) We conducted an ablation study. We directly use Eq 5 as the similarity metric to compare and select the most similar models to distribute back to the clients. With all the identical settings in the subsection "Large Number of Clients" of the original paper, the results are as below:
> | Public Data | Approach | IID     | Non-IID |
> |-------------|----------|---------|---------|
> | Labeled     | Ablation | 63.68\% | 56.09\% |
> |             | pFedHR   | 80.02\% | 77.63\% |
> | Unlabeled   | Ablation | 57.40\% | 51.17\% |
> |             | pFedHR   | 78.98\% | 75.77\% |
>
> (2) We observe that direct applying the similarity as the metric to re-distribute the models would cause the performance to decrease significantly, especially under the non-IID setting.
>
> `>>> W5` ***Client  capacity***
>
> `>>> WA5`
>
> (1) With the constraints we designed, the size of the candidate model pools and the depth of the candidate model have been restricted. In fact, we find that the size of the largest generated candidate model is usually smaller than that of the largest client model.
>
> (2) Our work is also flexible for restricting the maximum depth of the generated candidate model. In our future work, we would like to add more specific strategies to generate size-aware candidate models for the specific application domain, such as IoT.
>
> `>>> W6-9` ***Presentation, reproduction, and minors***
>
> `>>> WA6-9`
>
> We truly thank the reviewer for the suggestions. We will revise as the reviewer suggested.
>
> `>>> Q1` ***Homogeneous setting***
>
> `>>> QA1`
>
> Thanks for your question. In the homogeneous setting, it would still follow all the automatic steps without humans involved. It is still possible to obtain different models’ architecture. When all the models are selected as M4, we show one obtained model structure in the attached PDF file.
>
> `>>> Q2` ***Multiple runs***
>
> `>>> QA2`
>
> We have provided the 3-time run average results:
>
>
> | Public Data | Dataset | MNIST             |                  | SVHN               |                  | CIFAR-10         |                  |
> |-------------|---------|-------------------|------------------|--------------------|------------------|------------------|------------------|
> |             | Model   | IID               | Non-IID          | IID                | Non-IID          | IID              | Non-IID          |
> | Labeled     | FedMD   | 93.06$\pm$1.22 \% | 92.21$\pm$1.36\% | 81.86$\pm$1.17\%   | 77.93$\pm$1.29\% | 68.55$\pm$1.52\% | 64.07$\pm$1.42\% |
> |             | FedGH   | 94.33$\pm$1.45\%  | 92.78$\pm$1.50\% | 81.77$\pm$1.89\%   | 80.68$\pm$2.03\% | 72.04$\pm$1.79\% | 70.53$\pm$1.85\% |
> |             | pFedHR  | 94.86$\pm$1.02\%  | 94.25$\pm$0.79%  | 84.32 $\pm$0.88\%  | 83.08$\pm$1.02\% | 73.64$\pm$0.86\% | 72.22$\pm$1.32\% |
> | Unlabeled   | FedKEMF | 93.23$\pm$2.06\%  | 91.07$\pm$2.77\% | 80.03 $\pm$1.96\%  | 77.27$\pm$1.86\% | 67.33$\pm$2.30\% | 64.80$\pm$1.88\% |
> |             | FCCL    | 93.49$\pm$2.33\%  | 92.11$\pm$2.45\% | 82.65 $\pm$ 2.01\% | 78.40$\pm$1.88\% | 68.13$\pm$1.69\% | 66.06$\pm$1.90\% |
> |             | pFedHR  | 93.96$\pm$1.37\%  | 93.25$\pm$1.68\% | 83.77$\pm$ 1.42\%  | 81.35$\pm$1.35\% | 71.96$\pm$1.01\% | 68.31$\pm$1.46\% |

---

> > ### Comment · Reviewer_xdZD · 2023-08-10
> >
> > I thank the authors for their answers. I have still some questions/remarks about the issues I mentioned.
> >
> > * the method does not seem suited for large-scale cross-device settings where clients may be selected only a few times.
> >
> > Perhaps my original remark was not clear. Consider a google-like cross-device setting where clients only participate a single time; your method would not work because no knowledge transfer from  \hat w_t^n would be possible (client n never comes back). This is not the case for the personalized FL algorithms I mentioned in my review. Even if the client participates a few times, your algorithm should show significant performance deterioration. The 1/20 sampling rate is still very far from typical sampling rates in large scale cross-device settings. Perhaps the authors should acknowledge that their method is suited for cross-silo federated learning.
> >
> > * applicability of privacy-preserving techniques like secure aggregation.
> >
> > I understand the authors confirm that existing privacy-preserving techniques cannot be applied to their algorithm.
> >
> > * Computational overhead.
> >
> > Kairouz et al, Advances and Open Problems in Federated Learning, mentions that typical values of B are between 50 and 50000. What are typical values of M and what the size of stitching parts?
> >
> > * The candidate model can be more complex than the client's model.
> >
> > What are the constraints which make the candidate model almost always smaller than the client's model?
> >
> > * improvement w.r.t. state of the art.
> >
> > The new experiments confirm that one cannot conclude the improvement to be statistically significant for MNIST and SVHN.

---

> > > ### Author Response · Authors · 2023-08-12
> > > **Thank the Reviewer for the Constructive Comments**
> > >
> > > Dear reviewer,
> > >
> > > We do sincerely appreciate your constructive comments and questions. With all the respect, we provide our replies and hope they would address your concerns appropriately.
> > >
> > >
> > > `>>> Cross-device setting V.S. cross-silo setting`
> > >
> > > `>>> Answer:` Thanks for the insightful comments. The current model design is indeed not optimally suitable to the large-scale cross-device federated learning. Your invaluable input has enabled us to accurately define the scope of our contributions within the realm of cross-silo federated learning. In our final version, we will provide a comprehensive clarification of the cross-silo federated learning setting and expound upon the limitations of our work concerning the cross-device scenario. Your feedback greatly assists in enhancing the clarity and context of our research.
> > >
> > > `>>> Computational overhead`
> > >
> > > `>>> Answer:` Thanks for pointing out. In our current experimental setting, we set B as 10 when the number of clients is 100, aligning with the setting of existing work [1,2]. We can increase the value of B in our experiments, but it could lead to an increase of computational overhead on the server side, which is a limitation of our current model design. In our future work, we will explore new strategies to tackle this limitation and make the technique of heterogeneous model reassembly more efficient. Your insights serve as valuable guidance in refining our model.
> > >
> > > In response to your query about the parameter M, we would like to clarify that it is indeed a learned value in our model. In our experiments on the SVHN dataset with 100 clients and 0.1 active client ratio, we observed a dynamic range for M across multiple runs, spanning from 7 to 12. Regarding the size of stitching parts, we appreciate your interest in this aspect. As exemplified in Figure 4 of our original paper, we presented two generated candidate models. These models featured 4 stitching parts at time t and 5 stitching parts at time t', each corresponding to different depths. For each stitching part between two consecutive layers in the candidate model, we have a nonlinear activation function on top of a linear layer (sec 3.1.3). Notably, each stitching part involves only two parameters: $\mathbf{W}$ and $\mathbf{b}$, which dynamically adjust in dimensions based on the specific layers being stitched. Importantly, when compared to the parameter quantities present in the original models, the size of these parameters remains relatively modest.  Furthermore, we only tune the stitching parts with fixing all the other parameters in the candidate models at the server side, which allows us to effectively enhance model performance while maintaining computational efficiency.
> > >
> > > [1] McMahan, Brendan, et al. "Communication-efficient learning of deep networks from decentralized data." Artificial intelligence and statistics. PMLR, 2017.
> > >
> > > [2] T Dinh, Canh, Nguyen Tran, and Josh Nguyen. "Personalized federated learning with moreau envelopes." Advances in Neural Information Processing Systems 33 (2020): 21394-21405.
> > >
> > > `>>> Improvement of state-of-the-art`
> > >
> > > `>>> Answer:`We appreciate your comments. It's worth noting that in comparison to the CIFAR-10 dataset, the MNIST and SVHN datasets are relatively less complex. While the performance enhancements on these latter two datasets may not always yield statistically significant results, it's still possible to discern discernible improvements.
> > >
> > > We would like to emphasize that these outcomes are based on scenarios where both client data and public data are derived from the same dataset.
> > >
> > > Furthermore, we have conducted another experiment to specifically investigate the impact of public data (Figure 1 in the original paper). Interestingly, when the public dataset and client data originate from different datasets, the performance gap between the proposed model and the baselines is further magnified. This underscores the significance of our approach and its potential for even greater differentiation in such scenarios.

---

> > > > ### Comment · Reviewer_xdZD · 2023-08-19
> > > >
> > > > Thank for your response.

---

### Official Review · Reviewer_nvYR · 2023-07-04

**Soundness:** 3 good
**Presentation:** 3 good
**Contribution:** 3 good
**Rating:** 6
**Confidence:** 4

**Summary:**

The authors designed a method to train a model with FL when clients have heterogeneous model architectures. They designed a model reassembly technique that stitches together parts of DNNs. pFedHR also creates personalised models for each client without requiring server-side data or explicit human guidance.

**Strengths:**

- Heterogeneous FL is an important problem
- The paper is well-written and easy to understand
- The authors compared their approach with number of existing algorithms

**Weaknesses:**

- The reason to stitch together heterogeneous architectures is not very well motivated. There are many other approaches that aim to learn personalised model for a set of clients with heterogeneous capabilities. For example FjORD [https://arxiv.org/abs/2102.13451]  train a number of subnets using adaptive dropout, HeteroFL [https://arxiv.org/abs/2010.01264] is another model where a superset is trained and submodes are used to address heterogeneity in FL, FedRolex [https://arxiv.org/abs/2212.01548], or [https://arxiv.org/abs/2210.16105] all use weight sharing to address similar tasks.

The authors should discuss what is the main advantage/motivation of having heterogeneous architectures stitched together over other approaches where the computational complexity of a model can be scaled up/down, and possibly compare with some of the above methods.

- Most experiments were done with few clients (N=12 up to 100). This is very small and might be unrealistic in real-=world applications where we might have million of clients participating in FL. It would be great to show how things scale up.

**Questions:**

- Maybe motivate better why should we personalise the whole architecture and not just the model weights ?
- In most cases, it would make sense to generate a few models for different device Tiers (e.g., low-tier old devices, mid-tier, high-tier). It is unclear why automatically generating client-specific personalised models is a good idea (we will have to ensure and measure the quality thousand of different architectures in production).

---

> ### Author Rebuttal · Authors · 2023-08-09
>
> We truly thank you for the insightful comments and suggestions. We hope our responses can address your concerns.
>
> `>>> W1` ***Motivation of applying stitching technique***
>
> `>>> WA1`
>
> (1) Thanks for the suggestions. Our approach is motivated by the challenge that clients may have different model structures and work together to deliver personalized models to enhance local client performance. Given the real-world setting where the clients’ models are heterogeneous, it can be treated as the “bottom-to-top” corporation in federated learning. We also appreciate the related studies mentioned by the reviewer in the comment. We will discuss them as follows.
>
> (2) The models, FjORD, HeteroFL, FedRolex, and AsyncDrop, all have a strong constraint, i.e., the client models are required to be submodels of the global model. This assumption might be too strong in real-world scenarios. In our work, we release this strong assumption, and any models can be used as client models. We assemble and reassemble the models with our designed strategies to obtain our candidate models to enable heterogeneous model cooperation.
>
> (3) In our approach, the model reassembly needs to follow the designed rules in Eq.~(1) of the original paper. This can help restrict the size of the candidate model pool. We also only fine-tune the parameters of the stitching layers rather than the whole generated models. In our experiments, the stitching layer is a nonlinear activation function on top of a linear layer, which is not very deep. Finally, it can be updated in a parallel way on the server side.
>
> `>>> W2` ***Larger scale clients***
>
> `>>> WA2`
>
> (1) Thanks for your comments. Following FedAvg and pFedMe, we conduct part of our experiments with 100 clients. In our original paper, we test the setting with 12 and 100 clients, respectively. As the reviewer suggested, we conducted an experiment on 200 clients and set the active client ratio to be equal to $10\%$. All the other settings keep the same as the subsection "Large Number of Clients" in Section 4.2 of the original paper. The results are shown below.
>
> | Public Data | Client Number | IID     | Non-IID |
> |-------------|---------------|---------|---------|
> | Labeled     | 100           | 80.02\% | 77.63\% |
> |             | 200           | 77.01\% | 73.53\% |
> | Unlabeled   | 100           | 78.98\% | 75.77\% |
> |             | 200           | 74.34\% | 71.04\% |
>
> (2) Compared with the results shown in the original paper, the performance has a slight decrease. Given the same amount of training data but a larger number of clients,  each client will have less training data. This would cause the performance to decrease at a certain range.
>
> `>>> Q1` ***Model personalization***
>
> `>>> QA1`
>
> We appreciate your question. With the heterogeneous model structures, we propose an approach via model disassembly and assembly to enable them to cooperate together. After we obtain the candidate models, we select one model for each client as the personalized teacher to guide the local modal update based on the similarity defined in Eq.~(5) of the original paper. In our proposed work, enabling heterogeneous model cooperation leads to different model architectures, and similarity-based matching helps select the most personalized teacher model for each local client at the next communication round.
>
> `>>> Q2` ***Model generation***
>
> `>>> QA2`
>
> Thanks for your valuable suggestion. We would like to share our answers below:
>
> (1) Following our designed rules, we have restricted the size of the candidate model pool and the depth of the generated models.
>
> (2) Using our current framework, we are able to simplify our approach to obtain a few models for clients with several tiers, e.g., low, medium, and high, as the reviewer mentioned. This would work for a more practical and specific domain, e.g., IoT.  We do appreciate your suggestion.

---

> > ### Comment · Reviewer_nvYR · 2023-08-10
> >
> > Thanks for the detailed response.
> >
> > > The models, FjORD, HeteroFL, FedRolex, and AsyncDrop, all have a strong constraint, i.e., the client models are required to be submodels of the global model. This assumption might be too strong in real-world scenarios.
> >
> > Can you elaborate on that ? why is this too strong ? What are the practical limitations that would make the above methods not feasible in the real world ?
> >
> > > Client number vs accuracy
> >
> > It seems the accuracy with higher number of clients is dropping. While each client has less data, the FL algorithm still goes through all the data in the dataset. So would you say this approach is scalable when we have million of possible clients in the real-world ?

---

> > > ### Author Response · Authors · 2023-08-11
> > > **Thanks for your prompt and insightful feedback.**
> > >
> > > Thanks for your valuable comments. We hope our answers could help address the questions appropriately.
> > >
> > > `>>> C1`
> > >
> > > To our best understanding, the major goal of distributing submodels to clients is to train a shared and accurate global model collaboratively in a “top-down” manner. Such approaches are efficient for large-scale training, but they require that all the client models must be extracted from the same global model.
> > >
> > > In our setting, we focus on the “bottom-up” scenario to train a personalized model for each client. In other words, our goal is different from that of the “top-down” manner. In such a way, we do not need to require all the client's models to be extracted from the same global model, and all clients can have different network structures. Hence, the strict requirement of client models being extracted from a shared large model might be somewhat impractical in real-world applications, especially when compared to our model's design.
> > >
> > > It is important to acknowledge the efficiency and effectiveness of submodel-based federated learning models in large-scale training, particularly in scenarios where a shared global model is desired.
> > >
> > >
> > > `>>> C2`
> > >
> > > In the case of specific datasets like CIFAR-10, the data quantity remains constant. As the number of clients increases, the amount of distributed data per client decreases. Although federated learning (FL) iterates over all available data, clients working with limited datasets might not achieve the same level of performance as those with ample training data. Consequently, this dynamic can lead to a reduction in the overall performance.
> > >
> > > Recognizing the limitations, we acknowledge that our existing design might not be ideally suited for expansive cross-device federated learning scenarios. Nevertheless, it does exhibit effectiveness within the cross-silo setting. In our forthcoming version, we will explicitly define and delineate this cross-silo setting, along with an open discussion of the scalability challenges inherent in our current model. Your feedback significantly contributes to the refinement of our work.

---

### Official Review · Reviewer_nXSK · 2023-07-07

**Soundness:** 3 good
**Presentation:** 3 good
**Contribution:** 3 good
**Rating:** 8
**Confidence:** 5

**Summary:**

In this paper, the authors introduce a technique that tackles the challenge of enabling collaboration among client models with different network structures in federated learning. Unlike traditional knowledge distillation (KD)-based approaches, the proposed model involves dividing the heterogeneous models into distinct parts and subsequently reassembling them. This unique reassembling approach fosters model diversity, facilitating personalized local training at each client. The proposed technique alleviates the adverse impact of utilizing public datasets on the server. By providing experimental results, the authors establish the effectiveness of their proposed model.

**Strengths:**

- Overall, the paper is well-written and easy to follow. The motivations behind the research are clearly articulated, and the experimental results presented are both sufficient and convincing.

- The introduction of the novel reassembling technique to federated learning is a significant contribution. This approach opens up new avenues for achieving model personalization, which has the potential to inspire further exploration by researchers in the field.

- The authors' inclusion of an empirical analysis that explores the negative impact of utilizing heterogeneous public data on the server is a notable and valuable aspect of this work. Moreover, the proposed reassembling solution effectively mitigates this issue, providing a practical and effective resolution.

**Weaknesses:**

- In the experiments, the authors use four self-designed CNN-based models for saving computational resources. However, there are some lightweight models, such as MobileNets, used in previous FL work. Incorporating these lightweight models would enhance the comprehensiveness of the experimental evaluation and provide a broader perspective on the proposed approach.
- Figure 1 is a little hard to read, and I suggest the authors use one setting as an example to demonstrate the challenge of using heterogeneous public data and put other results in Section 4.4.
- Adding a readme file to the source code package would be a useful addition.

**Questions:**

no

**Limitations:**

The authers have provided some discussions.

---

> ### Author Rebuttal · Authors · 2023-08-09
>
> Thanks for the reviewer's valuable comments.
>
> `>>> W1` ***Utilization of other models***
>
> `>>> WA1`
>
> Thanks for your suggestion. We conduct experiments using MobileNetV1, MobileNetV2, and MobileNetV3 as our client models. For the skipping connection, we treat the block as a whole without assembly. We maintain all the settings the same as Table 2 in Section 4.2 of the original paper. We report the results as below. The experimental results show that our proposed approach maintains superior performance over other baselines. Compared with the previous results in the original paper, we even obtain better performance.
>
> | Public Data | Dataset | MNIST   |         | SVHN    |         | CIFAR-10 |         |
> |-------------|---------|---------|---------|---------|---------|----------|---------|
> |             | Model   | IID     | Non-IID | IID     | Non-IID | IID      | Non-IID |
> | Labeled     | FedMD   | 93.28\% | 92.77\%  | 83.21\% | 78.66\% | 69.37\%  | 66.88\% |
> |             | FedGH   | 94.09\% | 94.04\% | 82.11\% | 80.29\% | 72.85\%  | 70.87\% |
> |             | pFedHR  | 94.98\% | 94.16\% | 84.89\% | 83.12\% | 74.96\%  | 72.98\% |
> | Unlabeled   | FedKEMF | 93.12\% | 91.45\% | 81.36\% | 80.47\% | 68.51\%  | 67.06\% |
> |             | FCCL    | 93.60\% | 91.72\% | 82.84\% | 80.58\% | 68.90\%  | 66.85\% |
> |             | pFedHR  | 93.72\% | 93.68\% | 84.66\% | 81.76\% | 72.26\%  | 69.42\% |
>
> `>>> W2` ***Figure quality***
>
> `>>> WA2`
>
> Thanks for the comments. We will revise and improve the figure quality as suggested.
>
> `>>> W3` ***Code support***
>
> `>>> WA3`
>
> We will add README to the open-source codes after the work has been accepted.

---

### Official Review · Reviewer_nMvK · 2023-07-07

**Soundness:** 3 good
**Presentation:** 3 good
**Contribution:** 4 excellent
**Rating:** 7
**Confidence:** 4

**Summary:**

The authors design a novel approach to make it possible for clients equipped with different model structures to cooperate in the federated learning framework. Specifically, the server will reassemble models into different parts and assemble them together. After that, they propose a similarity-based approach to match the most fitted model with the clients’ models and distribute them back to the clients respectively. To emphasize the advantages over the KD-based approaches, the authors discuss different perspectives of using public datasets. Finally, they address the effectiveness of the algorithm using experiment results under the IID and non-IID setting compared with other baselines.

**Strengths:**

1. This paper presents a new way to achieve personalized federated learning using heterogeneous model reassembly, which is significantly different from existing work.
2. The authors aim to design a new model to alleviate the issue of performance drop caused by introducing public data on the server, especially when its data distribution is different from that of clients.
3. The authors conduct extensive experiments on different settings, including 12 clients, 100 clients, IID, non-IID, and public data with labels and without labels. The experimental results demonstrate the effectiveness of the proposed model.

**Weaknesses:**

1. The training of stitching layers is not significantly clear. Are they trained with the other parts of the networks or trained separately? The authors can provide a description of how the networks are trained after they are stitched together.
2. Since the generated candidates may change for different runs, are the results averaged by multiple runs?
3. The font size used in Figure 1 is too small.

**Questions:**

Please see the weaknesses above.

**Limitations:**

The authors discussed the limitations of the proposed work.

---

> ### Author Rebuttal · Authors · 2023-08-09
>
> Thanks for the reviewer's comments. The responses to the weaknesses are shown as below.
>
> `>>> W1` ***Stitching layer training***
>
> `>>> WA1`
>
> I do appreciate your suggestion. After we stitch the models with stitching layers, we freeze the parameters of the selected layers but only train the parameters of the stitching layers.
>
>
> `>>> W2` ***Multiple runs***
>
> `>>> WA2`
>
> We provide the average results and stand deviation of 3 runs as below. From the results, we observe that the average results are still better than those of the baselines and the variances of different runs are smaller as well.
>
> | Public Data | Dataset | MNIST             |                  | SVHN               |                  | CIFAR-10         |                  |
> |-------------|---------|-------------------|------------------|--------------------|------------------|------------------|------------------|
> |             | Model   | IID               | Non-IID          | IID                | Non-IID          | IID              | Non-IID          |
> | Labeled     | FedMD   | 93.06$\pm$1.22 \% | 92.21$\pm$1.36\% | 81.86$\pm$1.17\%   | 77.93$\pm$1.29\% | 68.55$\pm$1.52\% | 64.07$\pm$1.42\% |
> |             | FedGH   | 94.33$\pm$1.45\%  | 92.78$\pm$1.50\% | 81.77$\pm$1.89\%   | 80.68$\pm$2.03\% | 72.04$\pm$1.79\% | 70.53$\pm$1.85\% |
> |             | pFedHR  | 94.86$\pm$1.02\%  | 94.25$\pm$0.79%  | 84.32 $\pm$0.88\%  | 83.08$\pm$1.02\% | 73.64$\pm$0.86\% | 72.22$\pm$1.32\% |
> | Unlabeled   | FedKEMF | 93.23$\pm$2.06\%  | 91.07$\pm$2.77\% | 80.03 $\pm$1.96\%  | 77.27$\pm$1.86\% | 67.33$\pm$2.30\% | 64.80$\pm$1.88\% |
> |             | FCCL    | 93.49$\pm$2.33\%  | 92.11$\pm$2.45\% | 82.65 $\pm$ 2.01\% | 78.40$\pm$1.88\% | 68.13$\pm$1.69\% | 66.06$\pm$1.90\% |
> |             | pFedHR  | 93.96$\pm$1.37\%  | 93.25$\pm$1.68\% | 83.77$\pm$ 1.42\%  | 81.35$\pm$1.35\% | 71.96$\pm$1.01\% | 68.31$\pm$1.46\% |
>
>
>
> `>>> W3` ***Presentation***
>
> `>>> WA3`
>
> Thanks for your suggestion. We will improve the quality of Figure 1 in the original paper.

---

### Author Rebuttal · Authors · 2023-08-09

We truly appreciate the insightful questions. We provide responses to Q1 from Reviewer xdZD. We hope it could help adequately address your concerns.

In the pdf file, there is a figure for the generated model under the homogeneous setting. Thanks again for your question.

---

### Decision · Program_Chairs · 2023-09-21

**Decision:**

Accept (poster)

**Comment:**

This paper aims to address the model heterogeneity problem in federated learning. In particular, the authors proposed a novel framework named pFedHR, which leverages heterogeneous model reassembly to achieve personalized federated learning. Reviewers agreed that the proposed framework is novel, and the experiments results are convincing. The authors' rebuttal has also successfully addressed most of the concerns from reviewers. The authors are highly encouraged to incorporate the suggestions from reviewers to their final version.